# End-To-End Latent Variational Diffusion Models for Inverse Problems in High Energy Physics

**Alexander Shmakov**
Department of Computer Science
University of California Irvine
Irvine, CA 92697
ashmakov@uci.edu

**Kevin Greif**
Department of Physics and Astronomy
University of California Irvine
Irvine, CA 92697
kgreif@uci.edu

**Michael Fenton**
Department of Physics and Astronomy
University of California Irvine
Irvine, CA 92697
mjfenton@uci.edu

**Aishik Ghosh**
Department of Physics and Astronomy
University of California Irvine
Irvine, CA 92697

Physics Division
Lawrence Berkeley National Laboratory
Berkeley, CA 94720

**Pierre Baldi**
Department of Computer Science
University of California Irvine
Irvine, CA 92697
pfbaldi@uci.edu

**Daniel Whiteson**
Department of Physics and Astronomy
University of California Irvine
Irvine, CA 92697

## Abstract

High-energy collisions at the Large Hadron Collider (LHC) provide valuable insights into open questions in particle physics. However, detector effects must be corrected before measurements can be compared to certain theoretical predictions or measurements from other detectors. Methods to solve this *inverse problem* of mapping detector observations to theoretical quantities of the underlying collision are essential parts of many physics analyses at the LHC. We investigate and compare various generative deep learning methods to approximate this inverse mapping. We introduce a novel unified architecture, termed latent variational diffusion models, which combines the latent learning of cutting-edge generative art approaches with an end-to-end variational framework. We demonstrate the effectiveness of this approach for reconstructing global distributions of theoretical kinematic quantities, as well as for ensuring the adherence of the learned posterior distributions to known physics constraints. Our unified approach achieves a distribution-free distance to the truth of over 20 times smaller than non-latent state-of-the-art baseline and 3 times smaller than traditional latent diffusion models.

## 1 Introduction

Particle physics experiments at the Large Hadron Collider study the interactions of particles at high energies, which can reveal clues about the fundamental nature of matter and forces. However, the properties of particles which result from the collisions must be inferred from signals in the detectors which surround the collision. Though detectors are designed to reconstruct the properties

37th Conference on Neural Information Processing Systems (NeurIPS 2023).

of particles with high fidelity, no detector has perfect efficiency and resolution. A common strategy to account for these effects is *simulation-based inference* [1], in which the detector resolution and inefficiency are modeled by a simulator. Samples of simulated events can then be compared to observed data to perform inference on theoretical parameters. However, simulators with high fidelity are computationally expensive and not widely accessible outside of experimental collaborations.

An alternative approach is the reverse, mapping the observed detector signatures directly to the unobserved *truth-level* information. In a particle physics context, this procedure is referred to as "unfolding"[1]. In practice, the quantum mechanical nature of particle interactions makes the forward map from the true particle properties to observed data not one-to-one. As a result, there is no true inverse function which can map a given detector observation to a single point in the truth-level space. Such *inverse problems* are challenging, but unfolded data allows for direct comparisons with theoretical predictions and across experiments, without requiring access to detector simulation tools which may not be maintained long-term.

Unfolding methods such as Iterative D'Agostini [2], Singular Value Decomposition [3], and TUnfold [4] have seen frequent use by experimental collaborations like ATLAS [5] and CMS [6]. However, these techniques are limited to unfolding only a few dimensions, and require binning the data, which significantly constrains later use of the unfolded distributions. The application of machine learning techniques [7] has allowed for the development of un-binned unfolding with the capacity to handle higher-dimensional data. One approach is to use conditional generative models, which learn to sample from the truth-level distributions when conditioned on the detector-level data; examples include applications of generative adversarial networks [8, 9], invertible networks [10, 11], and variational auto-encoders [12]. An alternative approach uses classification models as density estimators which learn to correct imprecise truth-level distributions with re-weighting [13–15]. Generative methods naturally produce unweighted events, an advantage over classification methods which may generate very large weights or even fail if the original distributions do not sufficiently cover the entire support of the true distribution. However, generative models are not always guaranteed to produce samples which respect the important physical constraints of the original sample. While making important strides, none of these methods have cracked the ultimate goal, *full-event unfolding*, where the full high-dimensional detector-level observations are mapped to truth-level objects.

This paper introduces a novel generative unfolding method utilizing a diffusion model [16–18] to map detector to truth-level distributions. Diffusion models are a class of generative models which learn to approximate a reverse noise diffusion process and have proven successful in natural image generation [19, 20] and recently scientific applications such as molecular link design [21]. Diffusion models excel in learning high-dimensional probability distributions at higher fidelity than normalizing flows and without the adversarial min-max loss of GANs. In HEP, they have already found use for approximating calorimeter simulations [22–25]. Latent diffusion models (LDMs), a specific class of diffusion models, perform the denoising in an abstract latent space [26] and excel in image generation tasks. These latent embeddings are often pre-trained on secondary objectives, such as VAE reconstruction tasks or CLIP [27], to limit computational and memory requirements. We unify the abstract embedding space of latent diffusion with the recently formalized variational diffusion approach [28] to develop an end-to-end variational latent diffusion model (VLD) achieving state-of-the-art performance in complex HEP generative tasks.

## 2 Background

### 2.1 Unfolding

Let $f_{\text{det}}(y)$ be the distribution which governs an observed detector-level data set $y = \{y_i\}$. An unfolding method aims to sample from a pre-detector distribution $f_{\text{parton}}(x)$, where *parton* refers to an unobserved state of interest to physicists. $f_{\text{parton}}(x)$ is related to $f_{\text{det}}$ via convolution with a "response" function $p(y|x)$ over the possible true values $x$. The response function describes the decay of the initial, unstable particles into stable particles and their interaction with the detector.

$$f_{\text{det}}(y) = \int dx\, p(y|x) f_{\text{parton}}(x) \tag{1}$$

---

[1]In other fields, this kind of problem is often referred to as "deconvolution".

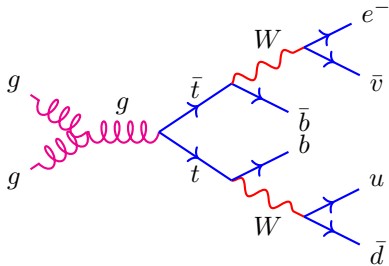

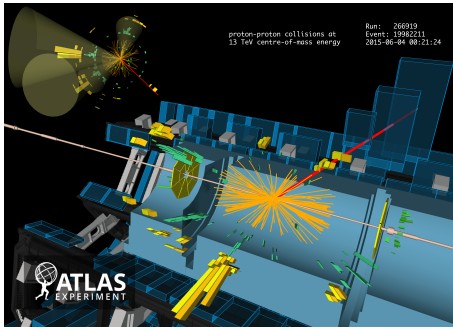

(a) Feynman diagram of top quark pair ($t\bar{t}$) production. Each top decays to a $W$ boson and a bottom (anti-)quark ($b$). In this example, one $W$ decays leptonically to an electron ($e^-$) and anti-neutrino ($\bar{\nu}$), the second decays hadronically to an up ($u$) and down ($\bar{d}$) anti-quark. We store the momentum of all final and intermediate particles.

(b) A example display of the high-dimensional detector observations for an LHC collision identified as likely to have contained a top quark pair. We show a 3D representation of the ATLAS detector and the hadronic jets which are produced by the decays. The detector-level data consists of the momentum measurements of these jets.

Figure 1: Visual representations of the different perspectives captured by the parton and detector level data. The parton-level data represents a fundamental theoretical description of the decay, whereas the detector-level data represents the real measurements observed after the decay. The primary challenge in unfolding is to infer the theoretical parton representation from the observed data.

No closed form expression exists for $p(y|x)$, but Monte-Carlo-based simulation can sample from parton values $x$ and produce the corresponding sample $y$. The parton distribution can be recovered via the corresponding inverse process if one has access to a pseudo-inversion of the response function $p(x|y)$, also known as the posterior.

$$f_{\text{parton}}(x) = \int dy \, p(x|y) f_{\text{det}}(y) \tag{2}$$

Generative unfolding methods build the posterior as a generative model, which can be used to sample from $p(x|y)$. The desired parton distribution is then obtained by Equation 2. Simulated pairs of parton-detector data, $(x, y)$, may be used to train the generative model.

An important issue when choosing to directly model the posterior is that this quantity is itself dependent on the desired distribution $f_{\text{parton}}(x)$, the prior in Bayes' theorem:

$$p(x|y) = \frac{p(y|x) f_{\text{parton}}(x)}{f_{\text{det}}(y)} \tag{3}$$

Producing the data set used to train the generative model requires choosing a specific $f_{\text{parton}}(x)$, which influences the learned posterior. In application to new datasets, this will lead to an unreliable estimate of the posterior density if the assumed prior is far enough from the truth distribution. A common method to overcome this challenge is to apply an iterative procedure, in which the assumed prior is re-weighted to match the approximation to the truth distribution provided by the unfolding algorithm [2]. Though application of this iterative procedure is not shown in this paper, the principle has been demonstrated with other generative unfolding methods [29], for which the conditions are similar.

## 2.2 Semi-Leptonic Top Quark Pair Production

Collisions at the LHC which result in a pair of top quarks allow for sensitive probes of new theories of physics, which makes measurement of the top quark properties an important task. Top quarks are unstable, decaying almost immediately to a $W$ boson and a bottom quark; the $W$ boson can then decay *hadronically* to two quarks or *leptonically* to a charged lepton and neutrino. The case where one of the produced top quarks decays hadronically and the other decays leptonically is known as the semi-leptonic decay mode, see Fig. 1a. The 4-momenta (three momentum components, one mass) of

these six objects (four quarks, the charged lepton, and the neutrino) constitute the parton-level space in this context.

The four quarks each produce a shower of particles (*jets*) which interact with the detector, while the neutrino passes through without leaving a trace. The resulting observed detector signature which defines the detector-level space is then quite complex, see Fig. 1b.

The semi-leptonic $t\bar{t}$ process has been studied by the ATLAS and CMS collaborations to measure various properties of the top quark and to search for new particles and interactions [30–35]. Many of these measurements use existing unfolding techniques, which limit the unfolded measurements to one or two dimensions. An un-binned and high dimensional unfolding technique would allow physicists to use the full power of their data.

## 2.3 Variational Autoencoders

Variational Autoencoders (VAEs) are a class of generative models combining an autoencoder architecture with probabilistic modeling [36, 37]. VAEs learn a non-linear latent representation of input data through an encoder and decoder network while incorporating probabilistic methods and sampling through the reparameterization trick [36]. VAEs have been applied to numerous applications, such as image synthesis [38] and natural language processing [39], among many others.

The VAE encoder network is parameterized as a probabilistic function, approximating the posterior distribution of the latent variables $z$ conditioned on the input data: $q(z|x)$. The decoder network likewise models the generative distribution conditioned on the latent variables $p(x|z)$. VAEs are trained by maximizing the evidence lower bound (ELBO), which is a lower bound on the log-likelihood of the data under the generative model [36]. The ELBO includes a reconstruction loss for training the decoder and a KL-divergence objective which enforces a regularization constraint on the learned latent posterior to a prior distribution $p(z)$.

$$\mathcal{L}_{\text{VAE}} = \mathbb{E}_{z \sim q(z|x)} \left[ -\log p(x|z) + D_{KL}(q(z|x) \parallel p(z)) \right] \tag{4}$$

Conditional VAEs (CVAEs) [40] extend the VAE framework by conditioning both the encoder and decoder networks on additional information, such as class labels, via an arbitrary conditioning vector $y$. This allows CVAEs to generate samples with specific desired properties, providing more control over the generated outputs.

$$\mathcal{L}_{\text{CVAE}} = \mathbb{E}_{z \sim q(z|x,y)} \left[ -\log p(x|z, y) + D_{KL}(q(z|x,y) \parallel p(z|y)) \right] \tag{5}$$

## 2.4 Variational Diffusion Models

Variational Diffusion Models (VDMs) define a conditional probabilistic generative model which exploits the properties of diffusion probabilistic models to generate samples by learning to reverse a stochastic flow [41]. VDMs may be seen as an extension of VAEs to a (possibly infinitely) deep hierarchical setting. The Gaussian diffusion process defines the forward stochastic flow with respect to time $t \in [0, 1]$ over the latent space $z_t \in \mathbb{Z}$ and conditioned on $y$ as:

$$q(z_t|x, y) \sim \mathcal{N}(\alpha_t x, \sigma_t \mathbb{I}) \tag{6}$$

The flow parameters, $\sigma_t$ and $\alpha_t$ are defined by a *noise schedule*. We use the continuous Variance Preserving (VP) framework throughout this work and derive these flow parameters based on a learned signal-to-noise ratio, $e^{-\gamma_\phi(t)}$, where:

$$\sigma_t = \sqrt{\text{sigmoid}(\gamma_\phi(t))} \text{ and } \alpha_t = \sqrt{\text{sigmoid}(-\gamma_\phi(t))}$$

Assuming it is possible to sample from the terminal distribution $p(z_1)$, we may produce samples from the data distribution by inverting the flow and sampling previous latent representations conditioned on future latent vectors. The inverse flow is modeled as $q(z_s|z_t, \hat{x}_\theta(z_t, t, y))$ where $\hat{x}_\theta$ is an approximate denoising of the original data at the current time-step. In practice, the data denoising is implemented using a variance-independent *noise prediction network*, $\hat{\epsilon}_\theta$, by the equation $\hat{x}_\theta(z_t, t, y) = \frac{(z_t - \sigma_t \hat{\epsilon}_\theta(z_t, t, y))}{\alpha_t}$. The noise prediction network, $\hat{\epsilon}_\theta$, is parameterized using a deep neural network. The learnable noise schedule $\gamma_\phi(t)$ is also parameterized using a positive valued, monotonic neural network with learnable end-points $\gamma_{min} = \gamma(0)$ and $\gamma_{max} = \gamma(1)$ [41]. Following

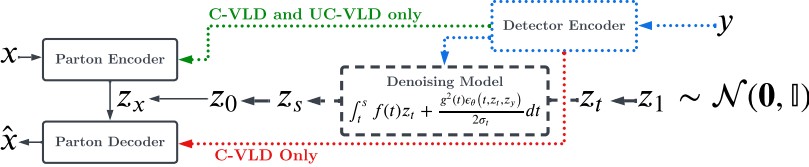

Figure 2: A block diagram of the end-to-end VLD models with trainable components. The conditional paths are drawn in different colors depending on which model variations employ them. We use the continuous, variance preserving SDE diffusion formulation introduced in [17] and [41]. We show the equivalent ODE form of SDE equation in the diagram.

the VP framework, the noise schedule is regularized so that the terminal distribution is the unit Gaussian: $p(z_1) \sim \mathcal{N}(\mathbf{0}, \mathbb{I})$. Both the noise prediction network and the noise schedule network are trained using the modified ELBO for continuous-time diffusion models [41]:

$$\mathcal{L}_{\text{VDM}} = D_{KL}(q(z_1|x, c) \parallel p(z_1)) + \mathbb{E}_{q(z_0|x)}\left[-\log p(x|z_0, y)\right]$$
$$+ \mathbb{E}_{\epsilon \sim \mathcal{N}(\mathbf{0}, \mathbb{I}), t \sim \mathcal{U}(0,1)}\left[\gamma'_\phi(t) \|\epsilon - \hat{\epsilon}_\theta(z_t, t, y)\|_2^2\right] \tag{7}$$

## 2.5 Latent Diffusion

Latent diffusion models (LDMs)[26] are a deep generative framework that operate the diffusion process in an abstract latent space learned by a VAE to sample high-dimensional data $p_D(x|z, y)$, possibly conditioned on a secondary dataset $p_C(y)$. This approach has proven dramatically successful when employed in natural image generation applications, including text-to-image synthesis, inpainting, denoising, and style transfer [26, 20].

LDMs first train an unconditional VAE to embed the data distribution into a low dimensional latent representation using a traditional VAE approach, $q(z_x|x)$ and $p(x|z_x)$, regularizing the latent space towards a standard normal $p(z_x) \sim \mathcal{N}(\mathbf{0}, \mathbb{I})$. A secondary encoder may be trained on the conditioning data $p(z_y|y)$ along-side VAE, typically using a CLIP objective [27] to map the two datasets into a common latent space. The diffusion process is then trained to reconstruct the latents $z_x$ from the flow latents $p(z_x|z_0, z_y)$. The diffusion model training remains otherwise identical to the standard diffusion framework.

Critically, the most successful methods train the VAE, the conditional encoder, and the diffusion process individually. While computationally efficient, this independence limits the models' generative power as each component is trained on subsets of the overall conditional generative objective. It may be possible to recover additional fidelity by instead training all components using a unified conditional generation objective. While several methods allow for training a VAE *along-side* diffusion [42, 43], these approaches either cannot train diffusion in the latent space or cannot account for a conditional, fully variational model. We construct a unified variational framework to allow for a conditional, probabilistic, end-to-end diffusion model.

## 3 Variational Latent Diffusion

This work integrates the learning capabilities of latent diffusion models with the theoretical framework of variational diffusion models in a unified conditional variational approach. This unified variational model combines the conditioning encoder, data VAE, and diffusion process into a single loss function. This framework enables further enhancement of these methods through a conditional data encoder or decoder, and an auxiliary physics-informed consistency loss which may be enforced throughout the network. We refer to this combined method as Variational Latent Diffusion (VLD), see Fig 2. The primary contributions of this paper are to define this unified model and derive the appropriate loss function to train such a model.

**Conditioning (Detector) Encoder**   In traditional LDMs, the conditioning encoder, $p(z_y|y)$, is pretrained through an auxiliary loss term, such as CLIP [27], which aims to unify the latent space of the conditioning and data. While this approach is efficient, it may not be optimal: the encoder is trained

on one objective, and then repurposed to act as a conditioning encoder for a separate generative model. With the end-to-end framework, we simultaneously learn this encoder alongside other generative terms, enabling us to efficiently train a variable-length, high-dimensional encoder fine-tuned for the generative objective. In this work, we simplify this conditioning encoder by restricting it to a deterministic mapping, $z_y = f_\theta(y)$. This decision is based on prior work such as the CVAE which opts for a deterministic conditional encoder, as well as for simplicity as there is a lack of motivating benefits from a stochastic encoder.

**Data (Parton) VAE** The traditional LDM VAE is unconditional, as this allows it to be easily pre-trained and reused for different diffusion models. As we are training a unified conditional generative model in an end-to-end fashion, we have the option to extend the encoder and decoder with conditional probabilistic models: $q_{\text{C-VLD}}(z_x|x, z_y)$ and $p_{\text{C-VLD}}(x|z_x, z_y)$. We experiment with both a conditional and unconditional VAE. Additionally, we explore an intermediate method that uses a conditioned encoder to estimate the VAE posterior, $q_{\text{UC-VLD}}(z_x|x, z_y)$, but employs an unconditional decoder during generation $p_{\text{UC-VLD}}(z_x|x)$.

**VLD ELBO** We interpret the continuous VDM as an infinitely deep hierarchical VAE as presented by Kingma *et al.* [41]. This interpretation allows us to seamlessly integrate the VAE into a unified diffusion framework by incorporating the VAE as an additional component in the hierarchy. Consequently, the hierarchical variational ELBO incorporates an extra KL divergence term, which serves to regularize the encoder posterior distribution [44]. We combine this hierarchical objective with the denoising loss term derived in [41] to define a combined ELBO for the entire generative model.

$$\mathcal{L}_{VLD} = D_{KL}(q(z_1|x, z_y) \parallel p(z_1)) + \mathbb{E}_{q(z_x|x, z_y)}\left[-\log p(x|z_x, z_y)\right]$$
$$+ D_{KL}(q(z_x|x, z_y) \parallel p(z_x|z_0)) + \mathbb{E}_{\epsilon \sim \mathcal{N}(\mathbf{0}, \mathbb{I}), t \sim \mathcal{U}(0,1)}\left[\gamma'_\phi(t) \parallel \epsilon - \hat{\epsilon}_\theta(z_t, t, z_y)\parallel_2^2\right] \quad (8)$$

The additional KL term may be derived explicitly if we assume a Gaussian VAE and a Gaussian diffusion process. The posterior is parameterized using a learned Gaussian, as in a standard VAE: $q(z_x|x, z_y) \sim \mathcal{N}(\mu_\theta(x, z_y), \sigma_\theta(x, y))$. The prior can be reformulated using the definition of the forward flow from Equation 6. Employing the reparameterization trick, we can rewrite the expression of $z_0$ in terms of $z_x$ as $z_0 = \alpha_0 z_x + \sigma_0 \epsilon$, where $\epsilon \sim \mathcal{N}(\mathbf{0}, \mathbb{I})$. Solving this equation for $z_x$ yields another reparameterized Gaussian, which allows us to define the prior over $z_x$ as:

$$p(z_x|z_0) \sim \mathcal{N}\left(\frac{1}{\alpha_0}z_0, \frac{\sigma_0}{\alpha_0}\mathbb{I}\right) \quad (9)$$

**Physics-Informed Consistency Loss** Reconstructing the mass of truth-level physics objects is challenging due to their highly peaked, low-variance distributions. For certain particles like leptons, the mass distribution exhibits a two-valued delta distribution, while for light quarks, it is consistently set to zero. Predicting these distributions is more difficult than predicting the energy of truth-level physics objects, which have a broader range. In special relativity, the mass ($M$), energy ($M$), and momentum **p** of a particle are related by $c^4 M^2 = E^2 - (c \parallel \mathbf{p} \parallel)^2$. We operate in natural units with a unit speed of light $c = 1$. Forcing the predicted mass, energy, and momentum to satisfy this equality improves stability and accuracy by capturing this underlying physical relationship between these quantities. We introduce a consistency loss, $\mathcal{L}_C$, in addition to the regular reconstruction loss, weighted by a hyper-parameter $\lambda_C$. Similar physics-informed constraints have previously been used for generative models in HEP [45–48]. The consistency loss minimizes the mean absolute error (MAE) between the predicted mass term and the corresponding predicted energy and momentum terms, encouraging the model to learn a more physically consistent representation.

$$\mathcal{L}_C = \lambda_C \left| \hat{M}^2 - \left(\hat{E}^2 - \parallel\hat{p}\parallel^2\right) \right| \quad (10)$$

## 4 Unfolding Semi-Leptonic $t\bar{t}$ Events

Generative models can be trained to estimate a conditional density given any set of paired data. In the unfolding context, a Monte Carlo simulation can be used to generate pairs of events at detector and parton level. The density of parton level events $f_{\text{parton}}(x)$ can be taken as the data distribution, and the density of detector level events $f_{\text{det}}(y)$ can be taken as the conditioning distribution. A generative model can then be used to unfold a set of observed events to the corresponding parton level events with the following procedure:

1. Sample a parton configuration from the distribution governing the process of interest: $x \sim p_D(x)$. This can be done using a matrix element solver such as MADGRAPH [49].

2. Sample a possible detector observation $y \sim p_C(y|x)$ using the tools PYTHIA8 [50] and DELPHES [51], which simulate the interactions of particles in flight and the subsequent interactions with a detector.

3. Train a generative model to approximate the inverse distribution $p_\theta(x|y)$.

4. Produce new posterior samples for inference data with unknown parton configurations.

## 4.1 Generative Models

Multiple baseline generative models are assessed alongside the novel VLD approach, with the goal of investigating the impact of each VLD component, including the conditional VAE, the denoising model, and the variational aspects of the diffusion. Note that the network architectures of the VAEs, denoising networks, and detector encoders are identical where relevant.

**CVAE** A traditional conditional Variational Autoencoder [40] approach employing a conditional encoder and decoder. We use a Gaussian likelihood for the decoder and a standard normal prior for the encoder, following conventional practices for VAE models.

**CINN** A conditional Invertible Neural Network [52], which represents the latest deep learning approach that has demonstrated success in unfolding tasks. This model utilizes a conditional normalizing flow to train a mapping from a standard normal distribution to the parton distribution, conditioned on the detector variables. The normalizing flow incorporates an All-In-One architecture [53], following the hyperparameters detailed in the CINN paper [52], which combines a conditional affine layer with global affine and permutation transforms to create a powerful invertible block. In this work, the MMD objective defined in [52] is replaced with a MSE reconstruction objective and the physics-informed consistency loss, for fair comparison with other models.

**VDM** A Variational Diffusion Model (VDM) [41] that aims to denoise the parton vector directly. This model serves as a baseline for examining the impact of the VAE in latent diffusion approaches. The denoising model is trained using a Mean Squared Error loss against the generated noise.

**LDM** A Latent Diffusion Model (LDM) with a pre-trained VAE, popularized by recent achievements in text-to-image generative models [26]. The VAE is pre-trained using a Gaussian likelihood and a minimal prior weight ($10^{-4}$). This baseline is meant to highlight the importance of the unified end-to-end architecture as all other aspects of the network are identical to the proposed method.

**VLD, C-VLD, UC-VLD** These models are variations on the proposed unified Variational Latent Diffusion (VLD) architecture. They correspond to an unconditional VAE (VLD), a conditional encoder and decoder (C-VLD), or a conditional encoder with an unconditional decoder (UC-VLD).

## 4.2 Detector Encoder

All of the generative models are conditioned on detector observations, represented as a set of vectors for each jet and lepton in the event, as described in Section 5. Additionally, the missing transverse momentum (MET) from the neutrino is included as a fixed-size global variable. As there is no inherent ordering to these jets, it is crucial to use a permutation-invariant network architecture for the encoder. We use the jet transformer encoder from the SPANet (v2.1, BSD-3) [54] jet-parton reconstruction network to embed detector variables. This architecture leverages the permutation invariance of attention to contextually embed a set of momentum vectors. We extract the fixed-size event embedding vector from the central transformer, mapping the variable-length, unordered detector observations into a fixed-size real vector $E_C(y) = z_y \in \mathbb{R}^D$.

## 4.3 Parton Encoder-Decoder

For a given event topology, partons may be represented as a fixed-size vector storing the momentum four-vectors of each theoretical particle. We describe the detailed parton representation in Section 5,

|        | Wasserstein | Energy | K-S | $KL_{64}$ | $KL_{128}$ | $KL_{256}$ |
|--------|-------------|--------|-----|-----------|------------|------------|
| **VLD**    | 108.76  | 7.59   | 4.08  | **3.47** | **3.74** | **4.53** |
| **UC-VLD** | **73.56** | **6.35** | **3.41** | 5.77 | 7.10 | 8.48 |
| **C-VLD**  | 389.62  | 25.39  | 4.65  | 9.54  | 10.09 | 10.79 |
| LDM    | 402.32  | 24.09  | 5.91  | 14.71 | 16.34 | 17.92 |
| VDM    | 2478.35 | 181.35 | 17.14 | 29.28 | 32.29 | 35.60 |
| CVAE   | 484.56  | 32.29  | 6.37  | 7.79  | 9.17  | 10.60 |
| CINN   | 3009.08 | 185.13 | 15.74 | 28.55 | 30.19 | 32.37 |

Table 1: Total distance measures across all 55 components for every model and metric. The independent sum of 1-dimensional distances for each component are summed across all the components to compute the total metrics.

which consists of a single 55-dimensional vector for each event. The encoder and decoder network employ a ConvNeXt-inspired block structure [55] for the hidden layers, described in Appendix A, which allows for complex non-linear mappings into the latent space. Unlike traditional VAE applications, our latent space may be *higher* dimensionality than the original space. The VAE's primary purpose therefore differs from typical compression applications, and instead solely transforms the partons into an optimized representation for generation.

The encoder uses this feed-forward block network and produces two outputs: the mean, $\mu_\theta(x, z_y)$, and log-standard deviation, $\sigma_\theta(x, z_y)$, of the encoded vector, possibly conditioned on the detector observations $z_y$. The decoder similarly accepts a latent parton representation, possible conditioned on the detector, and produces a deterministic estimate of the original parton configuration $\hat{x} = D(z_x, z_y)$.

## 5 Experiments

**Dataset** Each of the generative approaches is trained to unfold a simulated semi-leptonic $t\bar{t}$ production data set. Matrix elements are evaluated at a center-of-mass energy of $\sqrt{s} = 13$ TeV using MADGRAPH_AMC@NLO [49] (v2.7.2, NCSA license) with a top mass of $m_t = 173$ GeV. The parton showering and hadronization are simulated with PYTHIA8 [50] (v8.2, GPL-2), and the detector response is simulated with DELPHES [56] (v3.4.1, GPL-3) using the default CMS detector card. The top quarks each decay to a $W$-boson and $b$-quark, with the $W$-bosons subsequently decaying either to a pair of light $(u, d, s, c)$ quarks $qq'$ or a lepton-neutrino pair $\ell\nu$ $(\ell = e, \mu)$. A basic event selection is then applied on the reconstructed objects at detector-level. Electrons and muons are selected with a transverse momentum requirement of $p_T > 25$ GeV and absolute value of pseudorapidity $|\eta| < 2.5$. The $b$ and light quarks are reconstructed with the anti-$k_T$ jet algorithm [57] using a radius parameter $R = 0.5$ and the same $p_T$ and $|\eta|$ requirements as the leptons. Jets originating from $b$-quarks are identified with a "$b$-tagging" algorithm that incorporates a $p_T$ and angular $(\eta, \phi)$ dependent identification efficiency and mis-tagging rate. Selected events are then required to contain exactly one lepton and at least 4 jets, of which at least two must be $b$-tagged. Events are separated into training and testing data sets, consisting of 9,865,402 and 1,332,514 events respectively.

**Parton Data** The kinematics for the six final state partons are used as unfolding targets $(b, q_1, q_2, \bar{b}, \nu_l, l)$, along with the kinematics of the intermediate resonance particles $(W_{\text{lep}}, W_{\text{had}}, t, \bar{t})$, and the entire $t\bar{t}$ system. The parton-level data consists of 11 momentum vectors, each represented by the five quantities $(M, \log E, p_x, p_y, p_z)$; where $M$ is the invariant mass of the particle; $E$ is the energy; and $p_x$, $p_y$, and $p_z$ are the Cartesian coordinates of the momentum. The Cartesian components of the momentum are used for regression, as they have roughly Gaussian distributions compared to the commonly employed cylindrical coordinate representation. Although regressing both the mass and energy for each parton over-defines the 4-momentum, these components exhibit different reconstruction characteristics due to sharp peaks in the mass distributions. During evaluation, either the mass or energy can be used to compute any derived quantities. In our experiments, the regressed mass is only used for the mass reconstruction, and the predicted energy is used for other kinematics.

**Detector Observations** The detector-level jets and leptons are used as the conditioning data. The jets are stored as variable-length sets of momentum vectors with a maximum of 20 jets in each

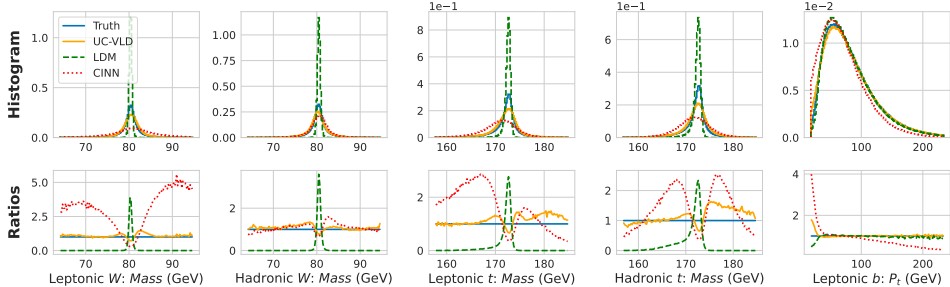

Figure 3: Highlighted reconstruction components. The top row presents the full global histogram while the lower plot presents the ratio between the predicted histogram and the truth. Notice the improved mass shape compared to the pre-trained and non-latent models.

event. This study is limited to semi-leptonic $t\bar{t}$ events, so each event is guaranteed to have a single lepton. The missing transverse momentum in each event (MET) is also computed and included in the conditioning. The jets and leptons are represented using both polar, $(M, p_T, \phi, \eta)$, and Cartesian, $(E, p_x, p_y, p_z)$, representations. We also include a one-hot particle identity, encoding either $\mu$ or $e$ for the lepton, or $b$ or non-$b$ for the jets as estimated by the $b$-tagger, resulting in 12 dimensions for each jet.

**Training**   Networks were trained using the MSE for the reconstruction and noise loss, along with the physics-informed consistency loss with a weight of $\lambda_C = 0.1$. Each model underwent training for 24 hours using four NVIDIA RTX 3090 GPUs, resulting in 500,000 to 1,000,000 gradient steps for each model. Models were trained until convergence and then fine-tuned with a smaller learning rate. Full hyperparameters are presented in Appendix B.

**Diffusion Sampling**   Variational diffusion models dynamically adapt the noise schedule during training by minimizing the variance of the ELBO [41]. After training, however, VDMs may employ a more traditional discrete noise schedule, and this approach is preferable when sampling for inference. The PNDM [58] sampler is used for generating parton predictions.

**Global Distributions**   Each trained model was evaluated on the testing data, sampling a single parton configuration for each detector-level event. The global distributions of the 55 reconstructed parton components were then compared to the true distributions. Complete unfolded distributions are presented in Appendix F. Several highlighted reconstruction distributions are presented in Figure 3. Additionally, each model was assessed using several distribution-free measures of distance. The bin-independent Wasserstein and Energy distances, the non-parametric Kolmogorov-Smirnov (K-S) test, as well as three different empirical KL divergence measures using 64, 128, and 256 bins, are presented in Table **??**. Full details about the distance functions are presented in Appendix C, and full tables of the distances per particle and per component are presented in Appendices D and E.

**Global Performance**   The two proposed VLD models with unconditional decoders (VLD and UC-VLD) consistently exhibited the best performance across all distance metrics. The end-to-end training procedure demonstrates improved performance over the pre-trained LDM model. It is interesting to note that UC-VLD has lower distribution-free distance wheras VLD has a lower histogram distance. This is likely because the histogram distance will soften the effect of outliers in the distribution as the bins will combine many different points into a single less noisy measurements. The conditional decoder in C-VLD and CVAE was found to worsen reconstruction. This is likely because the training procedure always employs the true encoded parton-detector pairs, $(z_x, z_c)$, whereas the inference procedure estimates the latent parton vector while using the true encoded detector variables for conditioning, $(\hat{z}_x, z_c)$. The lower performance may be evidence that this inference data technically falls out-of-distribution for the conditional decoder, indicating that an unconditional decoder is a more robust approach. Finally, we find that all latent models greatly outperformed the models that directly reconstructed the partons in data-space (CINN and VDM).

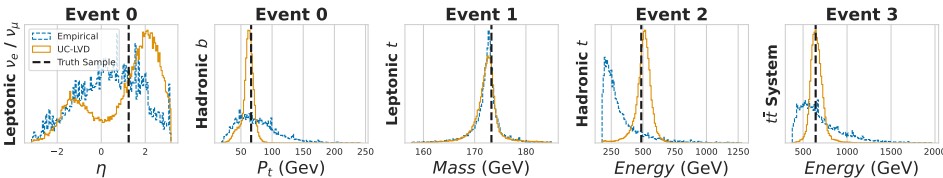

Figure 4: Highlighted reconstruction **per-event** posteriors for several events and components. We compare the LVD posteriors to an empirically brute-forced estimate of the posterior.

**Posterior Predictions** One key innovation of generative methods is the ability to sample from the posterior to illustrate the space of valid reconstructed partons for a detector level event. While the true posterior is not available, the unfolded distributions can be compared to a brute-force posterior distribution derived from the training data. This posterior is defined by a re-weighting of the parton level training data, where the weights are given by the inverse exponential of the $L_2$ distance between the testing event's detector configuration, $y_T$, and every training event's detector configuration, $y_i$: $w_i = e^{-\|y_T - y_i\|}$. Selected posterior distributions are presented in Figure 4, and complete posterior examples for individual events are presented in Appendix G. The latent diffusion models have much smoother posteriors than the empirical estimates, with the proposed VLD model producing more density close to the true parton configuration. Critically, the brute-force posterior often matches the unconditional parton level distribution, proving it is difficult to recover the true posterior. We also note that the VLD model was also able to reproduce a bimodal neutrino $\eta$ posterior. Neutrinos are not directly measurable at the detector, and their kinematics must be inferred from the event's missing energy. For events with a single neutrino, the missing neutrino energy defines a quadratic constraint on the term which often leads to two configurations satisfying both energy and momentum conservation. The network appears to learn this phenomenon and presents two likely $\eta$ values for the neutrino.

## 6 Conclusions

This paper introduced a novel extension to variational diffusion models, incorporating elements from latent diffusion models to construct a powerful end-to-end latent variational generative model. An array of generative models were used to unfold semi-leptonic $t\bar{t}$ events, an important inverse problem in high-energy physics. A unified model — combining latent representations, continuous variational diffusion, and detector conditioning — offered considerable advantages over the individual application of each technique. This addresses the challenge of scaling generative unfolding methods for high-dimensional inverse problems, an important step towards unfolding full collision events at particle-level. Despite being tested on a single topology, our method consistently improved baseline results, underscoring the importance of latent methods for such high-dimensional inverse problems. The framework presented may be broadly applicable to arbitrary topologies, although always limit to a single topology at a time. Future work will focus on broadening the method's applicability to different event topologies, unfolding to other stages of the event simulation chain (such as "particle level") to remove dependence on event topology, and evaluating its dependency on the simulator's prior distribution. The methods described in this study aim to provide a general end-to-end variational model applicable to numerous high-dimensional inverse problems in the physical sciences.

## 7 Acknowledgements

We would like to thank Ta-Wei Ho and Hideki Okawa for assistance in generating the $t\bar{t}$ sample used in this study. DW, KG, AG, and MF are supported by DOE grant DE-SC0009920, and AG is also supported under contract DE-AC02-05CH11231. The work of AS and PB in part supported by ARO grant 76649-CS to PB. We thank Vinicius Mikuni for fruitful discussions on unfolding methods.

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

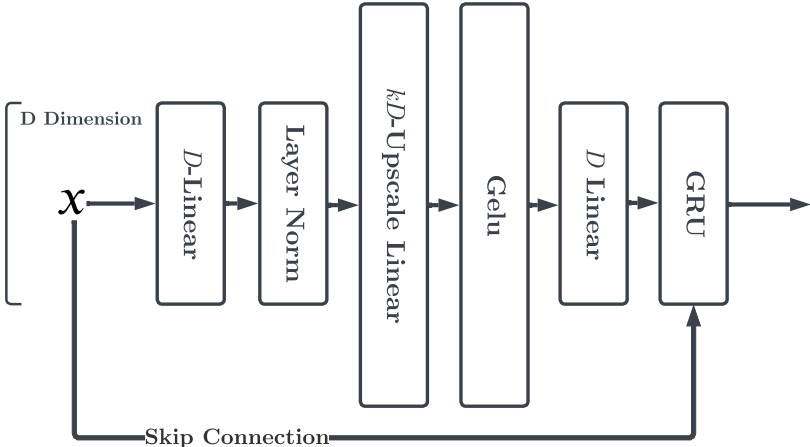

Figure 5: Block diagram from the gated inverse-bottleneck feed forward block used for the networks.

# A    Network Architecture

## A.1    Feed-Forward Block

A custom feed-forward block, derived from the successful ConvNeXt [55] and transformer models [59], is the foundation of the networks in this study. This block features an inverted bottleneck formed by three linear layers, a GELU activation [60], layer normalization [61] for regularization, and a gated residual connection inspired by GTRXL [62]. The block uses a GRU recurrent layer instead of a traditional skip connection, where the regular output is the input to the GRU and the skip-connection value is the hidden input. A complete block diagram of the feed-forward block is presented in Figure 5.

## A.2    Detector Encoder

The detector encoder network, as defined in SPANet [63, 54], processes variable-length sets of detector observations into a fixed-size $D$-dimensional vector. It uses the gated transformer architecture from SPANet version 2, with $N_S$ transformer encoder blocks. Instead of utilizing the tensor attention layers, an event-level representation of the detector observations is extracted from the central transformer encoder.

## A.3    Parton Encoder

The encoder starts with an embedding layer, transforming the fixed-size 55 dimensional parton representation into a $D$-dimensional vector via a linear layer. The encoder's body comprises $N_E$ feed-forward blocks arranged in series. The input can be the embedded $D$-dimensional parton vector, or its concatenation with the $D$-dimensional encoded detector data. Two independent networks, each accepting identical input and sharing the same block structure, predict the mean, $\mu_\theta(x; z_y)$, and the log standard deviation, $\log \sigma_\theta(x; z_y)$, respectively. Normalizing and then scaling the mean also helped prevent the encoder from learning very small-valued components: $\frac{\sqrt{D}}{\|\mu_\theta(x;z_y)\|} \mu_\theta(x; z_y)$

## A.4    Parton Decoder

The decoder retains the encoder's linear block structure. As a deterministic decoder, it comprises a single stack of $N_D$ feed-forward blocks and a concluding linear layer mapping $D$ dimensions back to 55. Conditional decoders also append the $D$-dimensional detector vector to the input before processing.

### A.5 Denoising Network

The denoising network, employing the same linear block structure, consists of $N_\epsilon$ feed-forward blocks. It maps the $D$-dimensional latent sample $z_t$ to the approximate noise that produced it, $\epsilon_\theta(z_t, t, z_y)$. The time, $t \in [0, 1]$, is encoded with a 32-dimensional sinusoidal position encoding as detailed in [59]. The latent vector, position encoding, and conditioning are concatenated and fed through the feed-forward network to produce the noise estimate.

## B Hyperparameters

We present a full table of hyperparameters used throughout the experiments. All models use the same set of parameters, ignoring any that do not apply to particle methods. Parameters were not tuned using rigorous search. The detector transformer parameters were extract from experiments presented in SPANet [54], and the other networks where tuned to contain a similar number of parameters as the detector encoder.

| Parameter | Value |
|---|---|
| Latent Dimensionality ($D$) | 96 |
| Attention Heads | 4 |
| Inverse Bottleneck Expansion ($k$) | 2 |
| Detector Transformer Encoder Layers ($N_S$) | 8 |
| Parton Encoder Blocks ($N_E$) | 6 |
| Parton Decoder Blocks ($N_D$) | 6 |
| Denoising Network Blocks ($N_\epsilon$) | 10 |
| Primary Learning Rate | $5 \cdot 10^{-4}$ |
| Fine-tuning Learning Rate | $1 \cdot 10^{-4}$ |
| $L_2$ Gradient Clipping Limit | 1.0 |
| Consistency Loss Scale ($\lambda_C$) | 0.1 |
| Batch Size (Per GPU) | 4096 |

Table 2: Table of complete hyperparameters used for training all generative models

## C Distance Metrics

As the parton global distributions do not have a known family of distributions to describe their components, model-free measures of distribution distance must be used to evaluate the models. Three different families of distance measures are used. These non-parametric distances are only defined for 1-dimensional distributions. As there is no commonly accepted way of measuring distance for $N$-dimensional distributions, the 1-dimensional distances are simply summed across the components. Although not ideal, it is enough to compare different models and rank them based on performance.

### C.1 Wasserstein Distance

The Wasserstein distance, often referred to as the earth-mover distance, quantifies the amount of work it takes to reshape one distribution into another. This concept originated from the field of optimal transport and has found wide applications in many areas, including machine learning. An equivalent definition defines this distance as the minimum cost to move and transform the mass of one distribution to match another distribution. For a pair of 1-dimensional distribution samples, denoted $u$ and $v$, the Wasserstein distance can be computed in a bin-independent manner. This is achieved by computing the integral of the absolute difference between their empirical cumulative distribution functions (CDFs), $U(x)$ and $V(x)$.

$$D_{\text{Wasserstein}}(u, v) = \int_{-\infty}^{\infty} |U(x) - V(x)| dx$$

### C.2 Energy Distance

Energy distance is another statistical measure used to quantify the difference between two probability distributions based on emperical CDFs. It compares the expected distance between random variables

drawn from the same distribution (intra-distribution) with the expected distance between random variables drawn from different distributions (inter-distribution). The Energy distance may be defined as the squared variant of the Wasserstein distance.

$$D_{\text{Energy}}(u, v) = \sqrt{2 \int_{-\infty}^{\infty} (U(x) - V(x))^2 \, dx}$$

### C.3 Kolmogorov-Smirnov Test

The two-sample Kolmogorov-Smirnov (K-S) test is a non-parametric statistical hypothesis test used to compare the underlying probability distributions of two independent samples. It is particularly useful in machine learning applications where the goal is to assess whether two datasets come from the same distribution or if they differ significantly, without making any assumptions about the underlying distribution shape. It is also based on empirical CDFs.

### C.4 KL-Divergence

An alternative approach to empirical CDF approaches is to bin the data into histograms and compute discrete distribution distances from these histograms. The common Kullback–Leibler distance is used with three different bin sizes. After finding the histograms with $N$ bins for $1 \leq i \leq N$, $P_N(i)$ and $Q_N(i)$, the discrete KL divergence is computed as

$$D_{KL,N} = \sum_{i=1}^{N} P_N(i) \log \left( \frac{P_N(i)}{Q_N(i)} \right)$$

## D Particle Distance Tables

Tables 3 to 6 present the distance metrics for each parton and model. The general trends in Table **??** remain generally consistent across partons. The neutrino reconstruction can prove difficult for Latent diffusion models, likely due to its very peaked components.

## E Component Distance Tables

Tables 7 to 10 present the distance metrics for each component and model. The mass component seems to vary the most for for many of the distance functions, indicating that many models struggle reconstructing the peaked mass distributions. However, the overall results remain consistent with Table **??**. We again see the clear benefit of both latent diffusion and end-to-end training.

## F Global Distribution Plots

Figures 6 through 16 present a collection of global distributions for the three primary classes of generative models for every particle and component. The proposed method (VLD) closely matches the truth distributions across all components, including the mass which is slightly smoothed but peaks in the correct location. Baseline models struggle with capturing the peaks and shapes of the distributions.

## G Posterior Distribution Plots

Figures 17 and 18 present a collection of posterior distributions for four testing events, along with several models and the brute-force empirical approach.

## H Loss Function Derivation

We provide a derivation for the loss function presented in Equation 8, adhering to the generative model displayed in Figure 2. Here, data is generated from a latent representation, $p(x|z_x)$; the latent

data from the diffusion end-point, $p(z_x|z_0)$; a diffusion process to generate the sampled configuration $p(z_0|z_1)$; and a simple standard normal distribution prior at the base of the hierarchy $p(z_1)$. All these functions are further conditioned on the (embedded) detector observations $z_y$. We start with the full conditional hierarchical VAE loss function over $L$ variational layers as per [64].

$$\mathcal{L}_{\text{ELBO}} = \mathbb{E}_{q(z_x|x,z_y)}\left[-\log p(x|z_x, z_y)\right] + \sum_{l=1}^{L} D_{KL}(q(z_l|x, z_y, z_{i \neq l}) \parallel p(z_l|z_{<l})) \quad (11)$$

Next, we substitute the layers we defined for the sum to expand the expression for our generative model. For this step, we employ a three-stage hierarchy with the following substitutions: $z_1 \leftarrow z_1$, $z_2 \leftarrow z_0$, and $z_3 \leftarrow z_x$. We derive each of these components in the following sections.

$$\mathcal{L}_{\text{ELBO}} = \mathbb{E}_{q(z_x|x,z_y)}\left[-\log p(x|z_x, z_y)\right] \quad (12)$$
$$+ D_{KL}(q(z_1|x, z_y, z_x) \parallel p(z_1)) \quad (13)$$
$$+ D_{KL}(q(z_0|x, z_y, z_1, z_x) \parallel p(z_0|z_1)) \quad (14)$$
$$+ D_{KL}(q(z_x|x, z_y, z_1, z_0) \parallel p(z_x|z_1, z_0)) \quad (15)$$

## H.1  Prior Loss

Equation 13 establishes the prior loss and the base layer in the hierarchy. In accordance with the VP framework, the correct prior distribution for the final latent representation is the standard normal $p(z_1) \sim \mathcal{N}(\mathbf{0}, \mathbb{I})$. We learn the noise schedule via $\log SNR(t) = -\gamma_\phi(t)$, as defined in VDM [41]. As such, we must ensure the terminal state in the forward diffusion process aligns with the prior distribution. Substituting the VP noise schedule yields the following distribution for the posterior: $q(z_1|x, z_y, z_x) = \mathcal{N}(\alpha_1 z_x, \sigma_1 \mathbb{I})$, where $\sigma_t = \sqrt{\sigma(amma_\phi(1))}$, and $\alpha_1 = \sqrt{\sigma(-\gamma_\phi(1))}$ as obtained from Section 2.4. We estimate this KL divergence through Monte-Carlo sampling:

$$D_{KL}(q(z_1|x, z_y) \parallel p(z_1)) = \mathbb{E}_{z_x \sim q(z_x|x,z_y)}\left[(\alpha_1 z_x)^2 + \sigma_1^2 - \log(\sigma_1^2) - 1\right] \quad (16)$$

## H.2  VAE Loss

Equation 15 delineates our contribution to this unified variational model. We derive the base distributions for the right-hand distribution in Equation 9: $p(z_x|z_0) \sim \mathcal{N}\left(\frac{1}{\alpha_0}z_0, \frac{\sigma_0}{\alpha_0}\mathbb{I}\right)$. We further describe that the posterior distribution in this KL term is merely the regular VAE posterior $q(z_x|x, z_y) \sim \mathcal{N}(\mu_\theta(x, z_y), \sigma_\theta(x, z_y))$. We define the posterior over $z_0$ given $z_x$ by adhering to the definition of the VP diffusion process $q(z_0|z_x) \sim \mathcal{N}(z_x, \sigma_0 \mathbb{I})$. As all these distributions are normal, we can provide an explicit form for this loss:

$$D_{KL}(q(z_x|x, z_y) \parallel p(z_x|z_1)) =$$
$$\mathbb{E}_{z_x \sim q(z_x|x,z_y), z_0 \sim q(z_0|z_x)}\left[D_{KL}\left(\mathcal{N}(\mu_\theta(x, z_y), \sigma_\theta(x, z_y) \parallel \mathcal{N}\left(\frac{1}{\alpha_0}z_0, \frac{\sigma_0}{\alpha_0}\mathbb{I}\right)\right)\right]$$

where the KL term is the regular normal distribution KL.

$$D_{KL}(\mathcal{N}(\mu_0, \sigma_0) \parallel \mathcal{N}(\mu_1, \sigma_1)) = \frac{1}{2}\left[\frac{\sigma_0^2}{\sigma_1^2} + \frac{(\mu_0 - \mu_1)^2}{\sigma_1^2} + \log\sigma_1^2 - \log\sigma_0^2 - 1\right]$$

## H.3  Diffusion Loss

Equation 14 defines the final diffusion loss term for the denoising network. We follow the derivation from Kingma et. al. [41] for a continuous-time diffusion process. The key insight is to interpret the diffusion process as infinitely deep hierarchical variation model. The VP framework defines intermediate steps as: $q(z_t|x, y) \sim \mathcal{N}(\alpha_t x, \sigma_t \mathbb{I})$. Following [41], we derive a Monte-Carlo estimate

of the integral loss for a noise prediction network:

$$D_{KL}(q(z_0|x, z_y, z_1, z_x) \parallel p(z_0|z_1)) = -\frac{1}{2}\mathbb{E}_{\epsilon \sim \mathcal{N}(\mathbf{0}, \mathbb{I})}\left[\int_0^1 SNR'(t) \left\| z_x - \hat{z_x}(z_t, z_y, t) \right\|_2^2 dt\right]$$

$$= -\frac{1}{2}\mathbb{E}_{t \sim \mathcal{U}(0,1), \epsilon \sim \mathcal{N}(\mathbf{0}, \mathbb{I})}\left[SNR'(t) \left\| z_x - \hat{z_x}(z_t, z_y, t) \right\|_2^2\right]$$

$$= \frac{1}{2}\mathbb{E}_{t \sim \mathcal{U}(0,1), \epsilon \sim \mathcal{N}(\mathbf{0}, \mathbb{I})}\left[\gamma'_\phi(t) \left\| \epsilon - \hat{\epsilon}_\theta(z_t, z_y, t) \right\|_2^2\right]$$

## H.4 Reconstruction Loss

Equation 12 is the final and simplest aspect of this model. Since we are regressing the parton components, we use a Normal decoder on the VAE and use a simple Mean Squared Error loss as the primary reconstruction loss.

$$\mathcal{L}_{MSE} = \left\| \text{DECODER}(z_x, z_y) - x \right\|_2^2$$

As we explain in the text, this MSE loss works well for most components but fails to capture the peaked nature of the mass term. Therefore, we add the physics consistency loss described by Equation 10 in order to assist with this mass reconstruction.

$$\mathcal{L}_C = \lambda_C \left| \hat{M}^2 - \left( \hat{E}^2 - \|\hat{p}\|^2 \right) \right|$$

The total reconstruction loss is simply the sum of these two components.

$$\mathbb{E}_{q(z_x|x, z_y)}\left[-\log p(x|z_x, z_y)\right] = \mathbb{E}_{q(z_x|x, z_y)}\left[\mathcal{L}_{MSE} + \mathcal{L}_C\right]$$

|  | VLD | UC-VLD | C-VLD | LDM | VDM | CVAE | CINN |
|---|---|---|---|---|---|---|---|
| Leptonic $b$ | 11.96 | 5.23 | 9.08 | 16.74 | 212.29 | 33.85 | 143.49 |
| Leptonic $\nu_e$ / $\nu_\mu$ | 10.33 | 7.40 | 36.69 | 29.63 | 117.37 | 36.95 | 149.22 |
| Leptonic $e$ / $\mu$ | 2.95 | 2.76 | 3.07 | 11.13 | 155.87 | 8.72 | 96.89 |
| Hadronic $b$ | 9.52 | 4.72 | 11.38 | 15.66 | 226.58 | 39.02 | 132.69 |
| Hadronic $q_1$ | 8.88 | 7.19 | 24.26 | 35.33 | 187.24 | 58.33 | 180.42 |
| Hadronic $q_2$ | 8.56 | 5.93 | 53.15 | 43.56 | 99.16 | 46.42 | 123.12 |
| Leptonic $W$ | 8.18 | 8.19 | 43.63 | 37.87 | 213.51 | 32.96 | 260.12 |
| Hadronic $W$ | 7.55 | 6.94 | 39.97 | 50.28 | 215.10 | 53.85 | 308.33 |
| Leptonic $t$ | 9.57 | 9.12 | 46.56 | 43.27 | 323.00 | 46.51 | 408.39 |
| Hadronic $t$ | 15.12 | 6.57 | 36.25 | 48.82 | 343.31 | 59.08 | 444.66 |
| $t\bar{t}$ System | 16.15 | 9.52 | 85.58 | 70.04 | 384.93 | 68.85 | 761.74 |

Table 3: **Particle Distance**: Wasserstein Distances

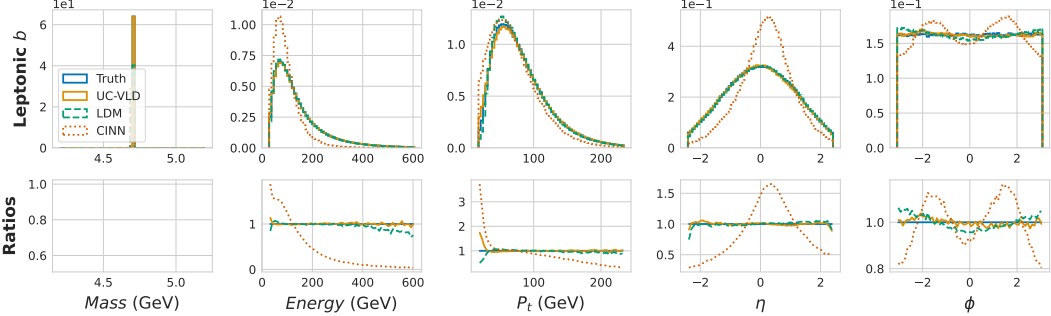

Figure 6: **Global Distribution**: Leptonic $b$ Quark

|  | VLD | UC-VLD | C-VLD | LDM | VDM | CVAE | CINN |
|---|---|---|---|---|---|---|---|
| Leptonic $b$ | 0.93 | 0.59 | 0.62 | 1.01 | 17.04 | 2.53 | 10.33 |
| Leptonic $\nu_e$ / $\nu_\mu$ | 0.79 | 0.73 | 2.31 | 1.76 | 11.02 | 2.72 | 12.15 |
| Leptonic $e$ / $\mu$ | 0.24 | 0.29 | 0.26 | 0.70 | 13.99 | 0.59 | 7.56 |
| Hadronic $b$ | 0.61 | 0.43 | 0.94 | 1.00 | 17.83 | 2.75 | 9.06 |
| Hadronic $q_1$ | 0.90 | 0.77 | 2.71 | 2.75 | 16.30 | 4.82 | 14.73 |
| Hadronic $q_2$ | 0.79 | 0.57 | 3.90 | 2.93 | 10.00 | 4.09 | 9.85 |
| Leptonic $W$ | 0.54 | 0.66 | 2.94 | 2.37 | 16.48 | 2.09 | 17.58 |
| Hadronic $W$ | 0.49 | 0.45 | 2.70 | 2.99 | 16.18 | 3.38 | 20.81 |
| Leptonic $t$ | 0.64 | 0.64 | 2.61 | 2.42 | 21.72 | 2.52 | 23.27 |
| Hadronic $t$ | 0.86 | 0.44 | 2.03 | 2.43 | 22.16 | 3.32 | 24.81 |
| $t\bar{t}$ System | 0.80 | 0.78 | 4.36 | 3.72 | 18.63 | 3.47 | 34.98 |

Table 4: **Particle Distance**: Energy Distances

|  | VLD | UC-VLD | C-VLD | LDM | VDM | CVAE | CINN |
|---|---|---|---|---|---|---|---|
| Leptonic $b$ | 0.85 | 0.07 | 0.06 | 0.59 | 1.91 | 1.01 | 1.30 |
| Leptonic $\nu_e$ / $\nu_\mu$ | 0.97 | 1.06 | 0.73 | 0.88 | 1.76 | 0.77 | 1.71 |
| Leptonic $e$ / $\mu$ | 0.48 | 0.33 | 0.52 | 0.35 | 1.47 | 0.43 | 1.02 |
| Hadronic $b$ | 0.05 | 0.05 | 0.10 | 1.06 | 1.95 | 0.90 | 1.30 |
| Hadronic $q_1$ | 0.40 | 0.40 | 0.60 | 0.63 | 1.63 | 0.81 | 1.50 |
| Hadronic $q_2$ | 0.90 | 1.00 | 0.83 | 0.76 | 1.64 | 0.95 | 1.59 |
| Leptonic $W$ | 0.10 | 0.11 | 0.46 | 0.37 | 1.39 | 0.30 | 1.27 |
| Hadronic $W$ | 0.07 | 0.09 | 0.42 | 0.42 | 1.36 | 0.32 | 1.46 |
| Leptonic $t$ | 0.09 | 0.10 | 0.35 | 0.32 | 1.61 | 0.30 | 1.43 |
| Hadronic $t$ | 0.10 | 0.09 | 0.32 | 0.28 | 1.60 | 0.37 | 1.48 |
| $t\bar{t}$ System | 0.06 | 0.09 | 0.26 | 0.24 | 0.82 | 0.21 | 1.68 |

Table 5: **Particle Distance**: Kolmogorov-Smirnov Test Statistics

|  | VLD | UC-VLD | C-VLD | LDM | VDM | CVAE | CINN |
|---|---|---|---|---|---|---|---|
| Leptonic $b$ | 1.50 | 0.01 | 0.01 | 0.49 | 5.91 | 0.27 | 1.44 |
| Leptonic $\nu_e$ / $\nu_\mu$ | 0.14 | 3.45 | 0.87 | 1.43 | 0.76 | 0.88 | 2.48 |
| Leptonic $e$ / $\mu$ | 0.14 | 1.35 | 0.44 | 0.93 | 3.66 | 1.04 | 3.43 |
| Hadronic $b$ | 0.00 | 0.01 | 0.01 | 3.57 | 5.99 | 0.29 | 1.09 |
| Hadronic $q_1$ | 0.34 | 2.07 | 0.46 | 1.19 | 3.80 | 1.72 | 6.15 |
| Hadronic $q_2$ | 1.54 | 0.08 | 0.67 | 0.96 | 2.31 | 0.98 | 2.67 |
| Leptonic $W$ | 0.02 | 0.03 | 2.94 | 2.59 | 2.39 | 0.69 | 2.13 |
| Hadronic $W$ | 0.01 | 0.02 | 2.43 | 2.54 | 2.11 | 1.25 | 2.53 |
| Leptonic $t$ | 0.02 | 0.03 | 1.22 | 1.47 | 2.50 | 0.98 | 2.24 |
| Hadronic $t$ | 0.03 | 0.04 | 0.99 | 1.14 | 2.39 | 1.05 | 2.45 |
| $t\bar{t}$ System | 0.01 | 0.01 | 0.06 | 0.04 | 0.48 | 0.04 | 3.59 |

Table 6: **Particle Distance**: KL Divergence with 128 bins.

|        | VLD   | UC-VLD | C-VLD  | LDM    | VDM    | CVAE   | CINN    |
|--------|-------|--------|--------|--------|--------|--------|---------|
| mass   | 2.06  | 2.61   | 17.76  | 19.12  | 87.24  | 20.91  | 113.29  |
| pt     | 7.18  | 7.65   | 39.39  | 33.44  | 429.70 | 74.56  | 257.20  |
| eta    | 0.37  | 0.26   | 0.47   | 0.40   | 0.44   | 0.51   | 4.35    |
| phi    | 0.22  | 0.19   | 0.20   | 0.38   | 0.17   | 0.30   | 1.29    |
| energy | 27.33 | 16.26  | 140.80 | 152.24 | 772.94 | 155.16 | 1098.99 |
| px     | 13.89 | 10.48  | 25.86  | 25.37  | 272.43 | 50.46  | 166.93  |
| py     | 8.90  | 7.95   | 26.48  | 22.95  | 270.28 | 47.97  | 168.37  |
| pz     | 48.80 | 28.16  | 138.66 | 148.42 | 645.15 | 134.69 | 1198.66 |

Table 7: **Component Distance**: Wasserstein Distances

|        | VLD  | UC-VLD | C-VLD | LDM  | VDM   | CVAE | CINN  |
|--------|------|--------|-------|------|-------|------|-------|
| mass   | 0.51 | 0.76   | 3.77  | 3.50 | 15.30 | 3.00 | 9.70  |
| pt     | 0.77 | 1.04   | 4.27  | 3.42 | 44.65 | 7.71 | 25.36 |
| eta    | 0.26 | 0.17   | 0.28  | 0.25 | 0.25  | 0.33 | 2.76  |
| phi    | 0.15 | 0.13   | 0.14  | 0.25 | 0.11  | 0.21 | 0.86  |
| energy | 1.40 | 0.96   | 7.50  | 7.36 | 53.11 | 8.32 | 70.01 |
| px     | 1.25 | 1.05   | 2.23  | 2.20 | 20.68 | 4.13 | 12.49 |
| py     | 0.87 | 0.81   | 2.23  | 1.88 | 20.54 | 3.90 | 13.15 |
| pz     | 2.38 | 1.43   | 4.97  | 5.23 | 26.71 | 4.70 | 50.79 |

Table 8: **Component Distance**: Energy Distances

|        | VLD  | UC-VLD | C-VLD | LDM  | VDM  | CVAE | CINN |
|--------|------|--------|-------|------|------|------|------|
| mass   | 3.39 | 2.78   | 3.20  | 4.54 | 7.55 | 4.21 | 4.66 |
| pt     | 0.08 | 0.14   | 0.36  | 0.31 | 3.30 | 0.67 | 1.92 |
| eta    | 0.14 | 0.09   | 0.13  | 0.12 | 0.11 | 0.18 | 1.34 |
| phi    | 0.07 | 0.06   | 0.07  | 0.12 | 0.05 | 0.10 | 0.39 |
| energy | 0.07 | 0.08   | 0.36  | 0.30 | 2.94 | 0.45 | 3.81 |
| px     | 0.10 | 0.10   | 0.18  | 0.18 | 1.16 | 0.31 | 0.78 |
| py     | 0.08 | 0.08   | 0.17  | 0.16 | 1.14 | 0.29 | 0.92 |
| pz     | 0.15 | 0.09   | 0.18  | 0.17 | 0.88 | 0.17 | 1.91 |

Table 9: **Component Distance**: Kolmogorov-Smirnov Test Statistics

|        | VLD  | UC-VLD | C-VLD | LDM   | VDM   | CVAE | CINN  |
|--------|------|--------|-------|-------|-------|------|-------|
| mass   | 3.69 | 7.04   | 9.78  | 15.80 | 22.78 | 8.53 | 12.34 |
| pt     | 0.01 | 0.02   | 0.07  | 0.07  | 3.08  | 0.22 | 1.66  |
| eta    | 0.01 | 0.01   | 0.03  | 0.04  | 0.04  | 0.03 | 1.87  |
| phi    | 0.00 | 0.00   | 0.00  | 0.01  | 0.00  | 0.01 | 0.07  |
| energy | 0.01 | 0.01   | 0.08  | 0.24  | 2.99  | 0.15 | 6.53  |
| px     | 0.01 | 0.01   | 0.03  | 0.03  | 1.28  | 0.10 | 0.92  |
| py     | 0.00 | 0.01   | 0.04  | 0.03  | 1.27  | 0.09 | 0.92  |
| pz     | 0.01 | 0.00   | 0.05  | 0.13  | 0.86  | 0.05 | 5.89  |

Table 10: **Component Distance**: KL Divergence with 128 bins.

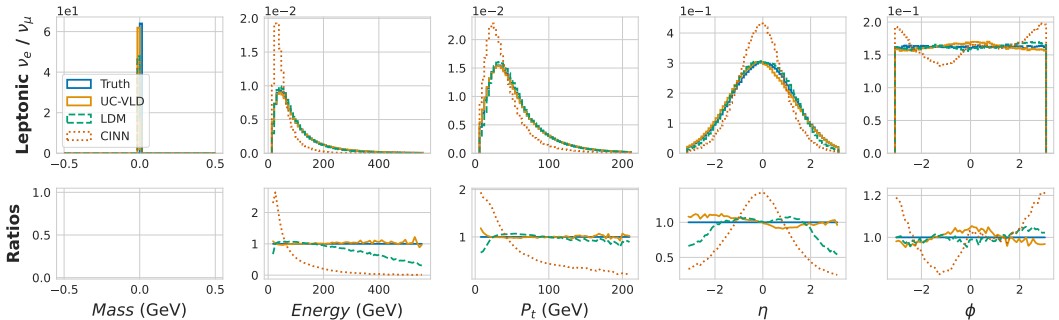

Figure 7: **Global Distribution**: Leptonic Neutrino $\nu_e$ / $\nu_\mu$

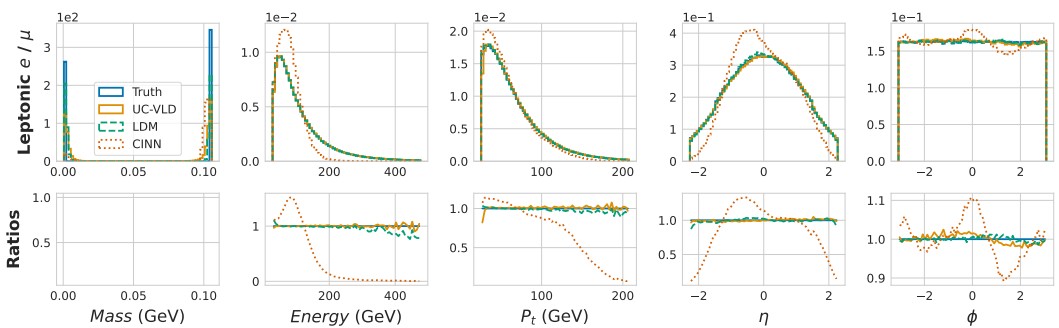

Figure 8: **Global Distribution**: Leptonic Lepton $e$ / $\mu$

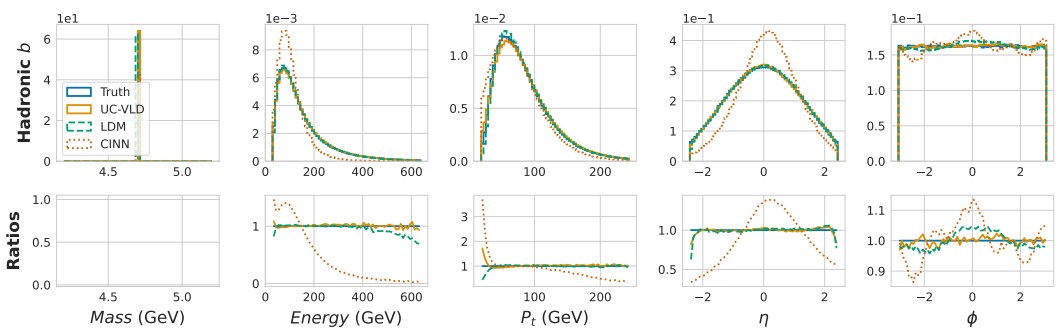

Figure 9: **Global Distribution**: Hadronic $b$ Quark

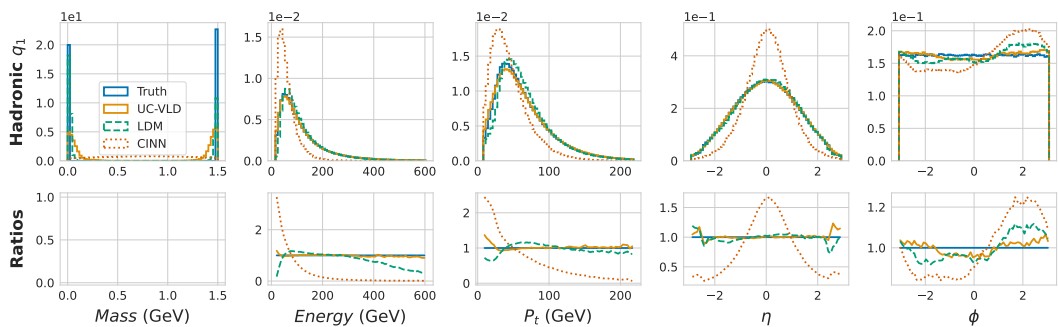

Figure 10: **Global Distribution**: Hadronic Light Quark

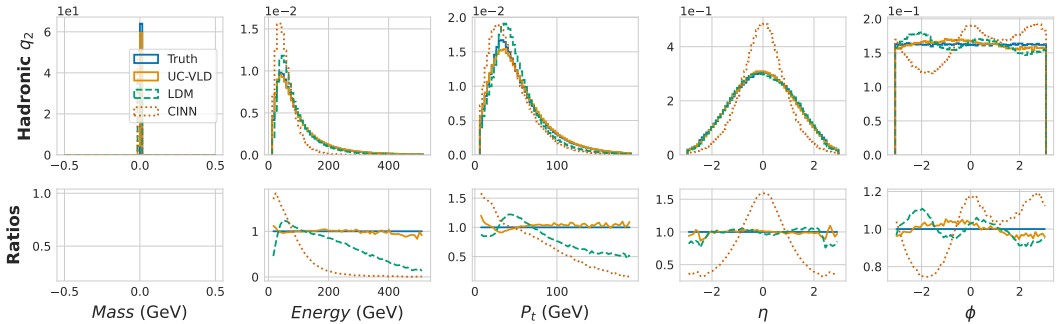

Figure 11: **Global Distribution**: Hadronic Light Quark

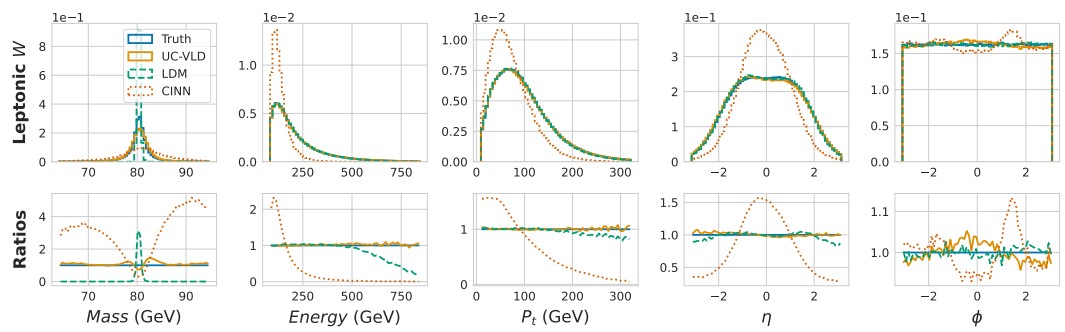

Figure 12: **Global Distribution**: Leptonic $W$ Boson

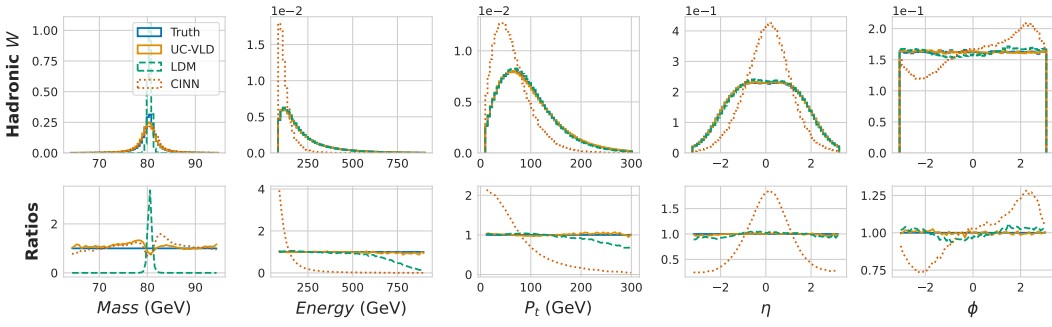

Figure 13: **Global Distribution**: Hadronic $W$ Boson

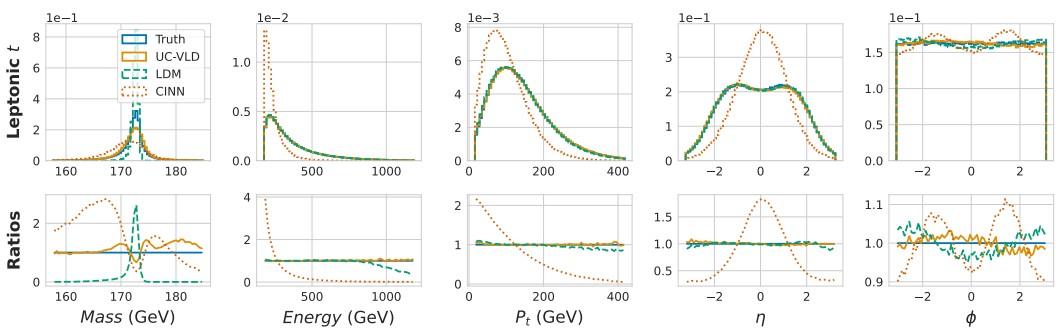

Figure 14: **Global Distribution**: Leptonic Top Quark

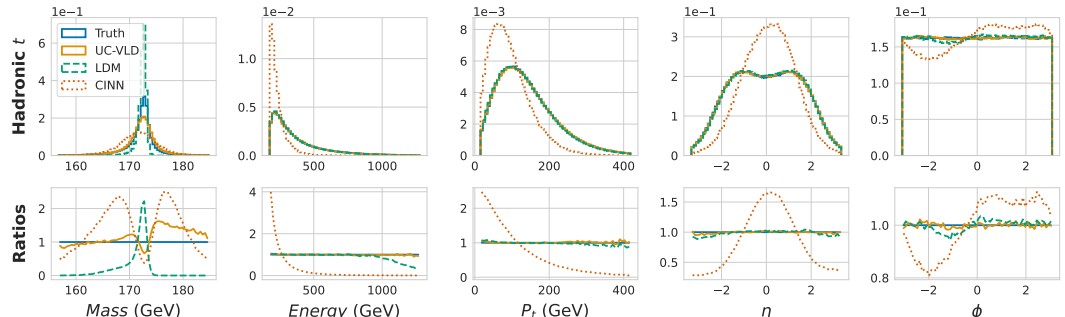

Figure 15: **Global Distribution**: Hadronic Top Quark

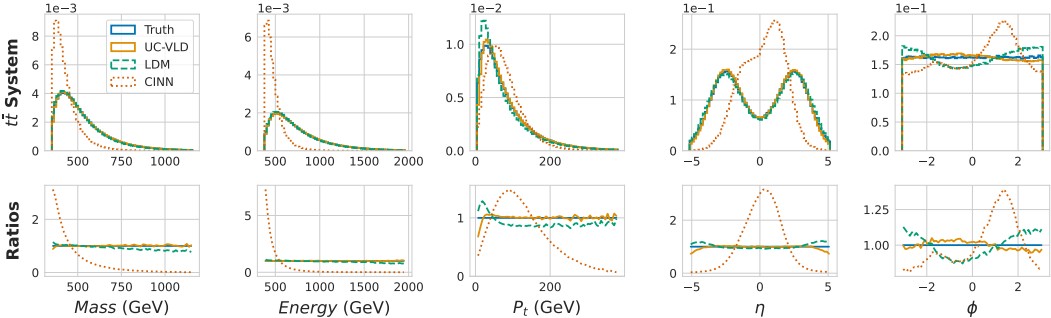

Figure 16: **Global Distribution**: Complete $t\bar{t}$ System

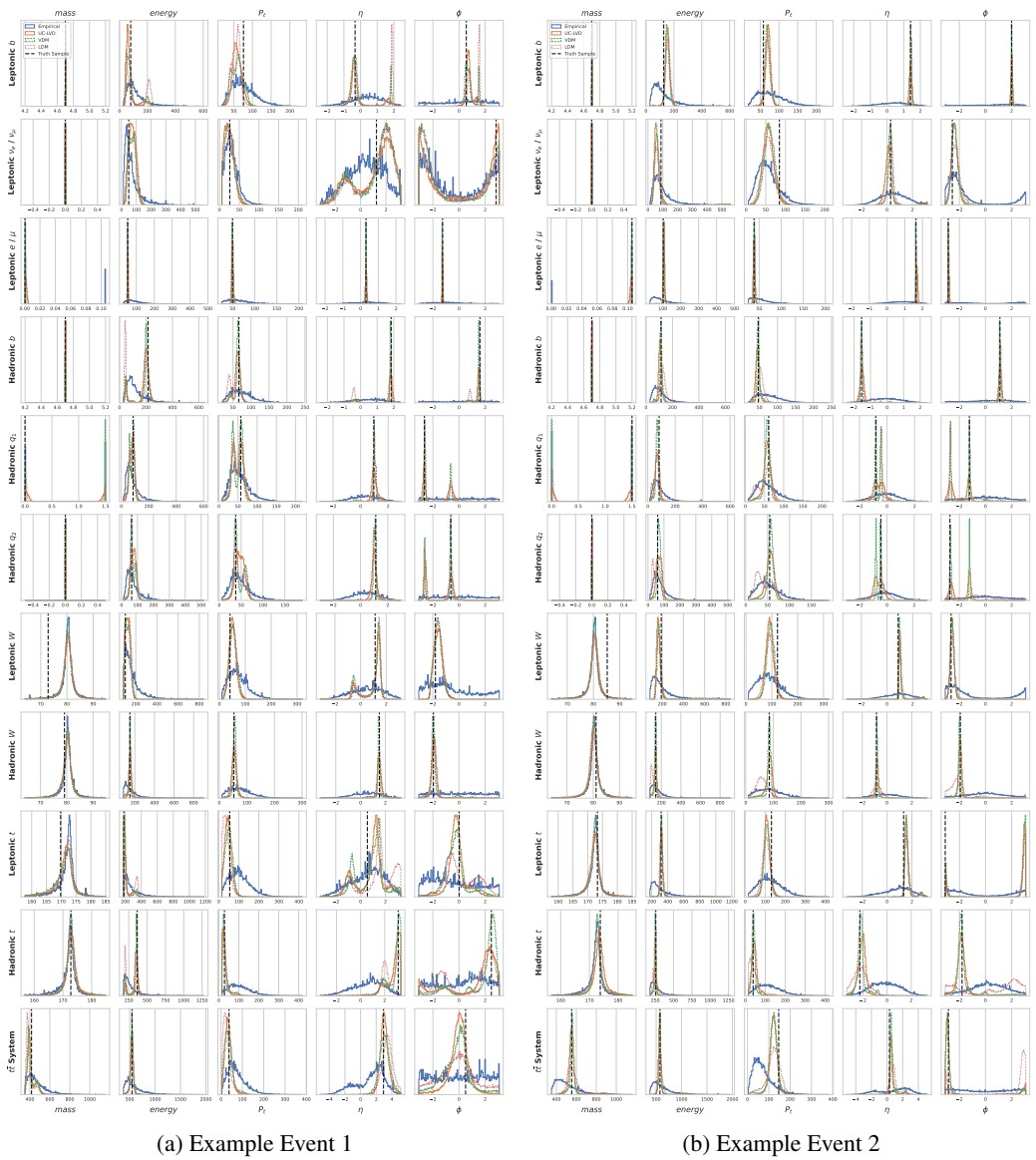

(a) Example Event 1       (b) Example Event 2

Figure 17: Posterior distributions for example events. Included is an empirical posterior distribution calculated from the training dataset.

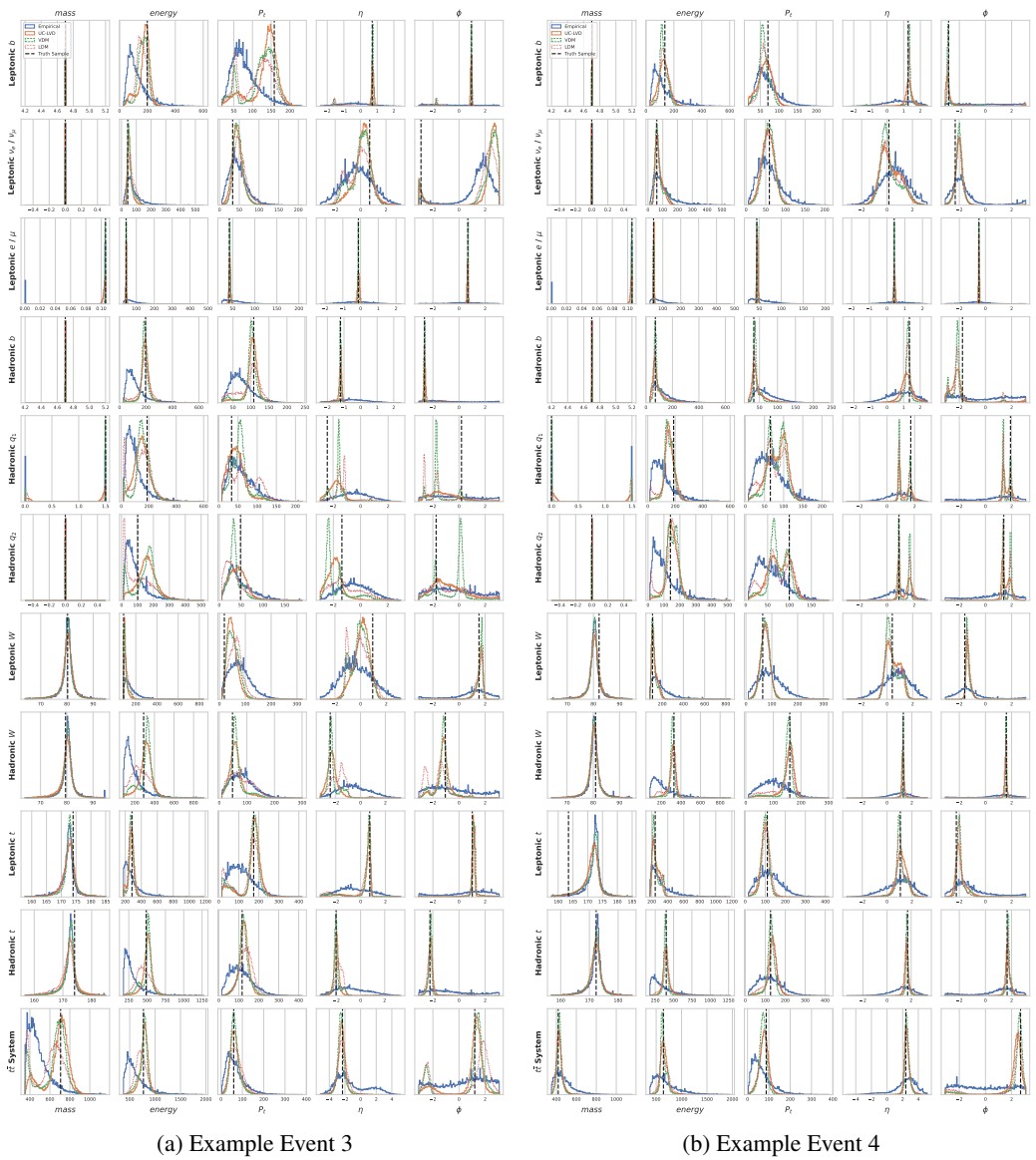

(a) Example Event 3          (b) Example Event 4

Figure 18: Posterior distributions for example events. Included is an empirical posterior distribution calculated from the training dataset.

