# OpenReview forum: "End-To-End Latent Variational Diffusion Models for Inverse Problems in High Energy Physics"
_NeurIPS.cc/2023/Conference — NeurIPS 2023 poster_

### Official Review · Reviewer_GAKk · 2023-06-25

**Soundness:** 3 good
**Presentation:** 4 excellent
**Contribution:** 3 good
**Rating:** 7
**Confidence:** 4

**Summary:**

The paper introduces a diffusion model based approach to tackle inverse problems. The method is then applied to High Energy Physics in order to reconstruct kinematic quantities.

**Strengths:**

The paper is well written with clear structure, definitions, and figures. The paper includes testing of the proposed methodology, VLD, on an important HEP inverse problem, which is unfolding semi-leptonic tt events. It is benchmarked on this task against other state of the art algorithms such as CINN, LDM, and VDM, as well  as variations of the authors own proposed algorithm. Their VLD algorithm does appear to significantly outperform the other considered algorithms.

Another strength of this paper is the fact that the authors are deliberate in construction of their network, explaining the advantages and necessity of each of the components.

**Weaknesses:**

Limited testing of the proposed unifying architecture is a weakness of the paper. Indeed, the authors do acknowledge that further testing on different event topologies would be beneficial, yet including at least one more benchmark would increase the confidence in the algorithm’s performance.

**Questions:**

1. Page 4, line 143, typo, should be “in an abstract”
2. Page 6, line 204, please define E to be energy, m mass, p momentum, and mention that you are using the HEP convention of setting c to be 1.
3. Please discuss the reason why UC-VLD outperforms the VLD on some metrics and vice-versa.
4. Figure 4, b quark, I don’t really see the bimodal nature of the distribution
5. Figure 4, neutrino, how is the neutrino “truth” line showing the bimodal nature?
6. What would happen with the LDM’s performance if a different prior weight is given?
7. What would happen in the benchmarks if the CINN uses the MMD loss instead?
8. To further show the improvement of the described unified training loss, it would be beneficial to show what would happen if the various components would be trained in parallel.

**Limitations:**

Yes

---

> ### Author Rebuttal · Authors · 2023-08-09
>
> We thank the reviewer for their king comments and detailed questions. We address the limitations comment in the global rebuttal. We answer the questions here.
>
> 1. & 2. We thank the reviewer for noticing these points. We have done a pass through the paper and fixed these presentation issues along with other small wording changes.
>
> &#8291;
>
> 3. We thank the reviewer for their insightful question. Unfortunately, we do not yet have an intuitive explanation for this behavior. The complexity is further compounded by the absence of a standard (and computationally feasible) method for measuring the distance between high-dimensional distributions. Each of the metrics we provide captures different aspects of distribution distance. We think the bin-independent metrics are more sensitive to outliers and long tails whereas the binned metrics focus more on the distance between the high-mass regions of the distribution. An examination of the detailed appendix results reveals that the UC-VLD performs more favorably with long-tailed features such as energy, but less so with the peaked mass term.
>
> 4. & 5. We regret the confusion surrounding the bimodal posteriors. Only the neutrino $\eta$ posterior is expected to be bimodal. This arises because neutrinos are not directly measurable at the detector, and their kinematics must be inferred from the event's missing energy. For events with a single neutrino, the missing neutrino energy defines a quadratic constraint on the $\eta$ term which often leads to two configurations satisfying both energy and momentum conservation. In Figure 4, we notice that the empirical estimate fails to capture this expectation, whereas the LVD captures this behavior without explicit information about this expectation. We also note that the truth sample will still be a single value sampled from this theoretically bimodal distribution. We have clarified this phenomenon in the revised text to better describe this expectation and our results.
>
> &#8291;
>
> 6. This is an interesting question. We use a low prior weight in accordance with the LDM paper, as we replicate their VAE pre-training. We suspect that, since our use case makes the latent dimension larger than the data dimension, the prior loss is dominating the pre-training step. This is because reconstruction loss may trivially be near-perfect without a bottleneck, so the main constraint for the VAE is the Gaussian prior over the latent space.
>
> 7.  The MMD (Maximum Mean Discrepancy) loss was used in the CINN paper to more accurately represent the true distributions associated with kinematics, such as the peaked mass term. We could not incorporate this MMD loss into our variational framework in a natural way as we employ a Gaussian VAE which necessitates a mean squared error loss term for the reconstruction. Instead, we chose to incorporate an additional self-consistent physics-informed regularization loss to enhance mass reconstruction, citing analogous strategies previously implemented for other generative models in physics. These two approaches share similar goals, although a detailed comparison of these methods could be fruitful for future work. We use exactly the same loss for all methods with the goal of evaluating only the generative model architecture's effect on performance,
>
> 8. We agree with this reviewer and in fact we reported the results of such an experiment in Table 1. In this table, we compare the component-wise training (LDM) to the unified end-to-end training (VLD), and show that the latter outperforms the former by roughly one order of magnitude. Since we use identical network architectures for all methods, the primary difference between these two methods is the unified training. We have further clarified this point in the revised version,

---

> > ### Comment · Reviewer_GAKk · 2023-08-10
> >
> > Dear authors,
> >
> > Thank you for the time taken to write the rebuttal and addressing the points I raised.

---

### Official Review · Reviewer_sYuc · 2023-06-28

**Soundness:** 3 good
**Presentation:** 3 good
**Contribution:** 2 fair
**Rating:** 4
**Confidence:** 4

**Summary:**

This paper proposes a unified framework, which combines the latent diffusion model and variational diffusion model, and applies the method to the inverse problem in the field of High Energy Physics. The loss terms for VAE and variational diffusion model are combined to achieve end-to-end training. The proposed VLD and corresponding variants achieve state-of-the-art performance as shown in experiments.

**Strengths:**

* A unified framework to support end-to-end training for latent variational diffusion model.
* The proposed VLD and variants achieve state-of-art performance existing generative models.


**Weaknesses:**

Overall, the main contribution of this paper is the unified framework to combine latent diffusion model and variational diffusion model, where the core lies in the extra VAE loss (3rd term in Equation 8).
1. Technical novelty seems limited, since the only contribution could be summarized as an extra loss term in diffusion model.
2. The experiments to emphasis the importance of the extra loss term are limited. Among all the baselines, the LDM at L243-245 is the most similar method to the proposed one, where the only difference is the extra loss term that leads to an end-to-end training process. The small gaps between LDM and C-VLD in Table 1 also suggest the similarities. The main question is, is the comparison between LDM and VLD fair enough? The details about training LDM are missing, such as whether they adopt the same feedforward block as VLD, especially how to pretrain the VAE and what is the performance of the pretained-VAE. Existing experiments seem insufficient to support that the benefits of VLD are from the unified training process.
3. Since there are results for C-VLD and UC-VLD, which mainly differ in the conditional signal, is it possible to compare LDM with similar settings?

**Questions:**

* How to define that the dataset at L273 includes enough variations for the high energy physics? For example, is there any existing similar settings for reference?

**Limitations:**

Technically, this paper provides a unified way, which seems like a straight-forward combination of the loss terms, to train a latent variational diffusion model in an end-to-end manner. However, experiment results are insufficient to support the importance of the extra loss term, or the benefits brought by the unified training process.

---

> ### Author Rebuttal · Authors · 2023-08-09
>
> We thank the reviewer for their insightful questions.
>
> 1. We respectfully disagree with this characterization. Many important contributions in machine learning, such as weight regularization and the original VAE, may be characterized in this reductive manner if one overlooks the reasoning behind the  respective loss modification. The primary contribution of this paper is a formal derivation of a unified variational architecture, together with its application to an important inverse problem in physics and an assessment of each component of this architecture. End-to-end training of this architecture in a theoretically justified fashion necessitates the addition of a new loss term, derived from the more foundational variational objective. Our experiments demonstrate that the main performance advantage originates from this unified end-to-end training of all the components, allowing the latent space to be fine-tuned to the diffusion task, in contrast to VAEs trained without diffusion or methods lacking a latent space.
>
> 2. We respectfully disagree with the reviewer's interpretation of the results. The VLD and LDM share identical network architectures for the denoising network, detector encoder, and VAE. The primary distinction between these two methods lies in pre-training the VAE (LDM) as opposed to training all components end-to-end (VLD). We have highlighted this distinction in the revised text to avoid confusion. Our findings, presented in Table 1, reveal that the regular VLD significantly outperforms the LDM, with the sole discrepancy being the unified training. We did observe that including a conditional decoder unexpectedly worsened performance, a phenomenon we discuss in the text in the paragraph starting on line 321. We hypothesize that this reduction may stem from the conditional decoder's sensitivity to minor errors in the diffusion process during inference.
>
> 3. We thank the reviewer for their interesting suggestion. While it may be possible to formulate a pre-trained LDM which uses a conditional VAE, doing so is non-trivial. The design would need to decide between using individual conditioning encoders for each phase or sharing the encoder for both phases. This is further complicated by the fact our domain lacks a pre-trained conditioning akin to CLIP for text. We encourage future work to examine this problem, but one of the benefits of our unified approach is that this change is simpler to implement while maintaining a common framework.

---

### Official Review · Reviewer_Btsg · 2023-07-05

**Soundness:** 4 excellent
**Presentation:** 4 excellent
**Contribution:** 4 excellent
**Rating:** 9
**Confidence:** 5

**Summary:**

The paper benchmarks several network architectures on a real-life problem in particle physics, i.e. the problem of inverting the effect of limited detector resolution and guess the features of a given collision from what is actually observed in the detector (unfolding). Considering several metrics to assess the accuracy of a given unfolding, the authors show that a novel architecture for diffusion models provides the best performance.

**Strengths:**

Very solid analysis, with clear explanation of the various steps.
Shows potential progress in applications, thanks to novel architecture

**Weaknesses:**

none

**Questions:**

none

---

> ### Author Rebuttal · Authors · 2023-08-09
>
> We thank the reviewer for their kind comments and support of our paper.

---

### Official Review · Reviewer_aaox · 2023-07-06

**Soundness:** 3 good
**Presentation:** 2 fair
**Contribution:** 2 fair
**Rating:** 4
**Confidence:** 4

**Summary:**

Thanks for the author's rebuttal. I have read all of the rebuttals and reviews and decided to keep the current rating.

Current work proposed an extension of the variational diffusion model, called the variational latent diffusion model. It is applied to inverse problems in the high energy physics field and tested on a single topology. The result shows that the distance is three times less than current latent diffusion models.

**Strengths:**

Originality: The model is a combination of two existing models, the latent diffusion model and the variational diffusion model. Moreover, the author defined an appropriate loss function to train the model.

Quality: The result demonstrated that the proposed model outperforms other baseline models.

Significance: The experiment provides unique data in the high energy physics field.

**Weaknesses:**

The paper has several typos and unclear points. Moreover, some claims are not well supported by the experiment results. The reviewer is confused to some presented points. Details can be found in the questions part.

**Questions:**

1. There is no general caption for Figure 1. Moreover, the sub-caption is not consistent with the sub-figure. For example, ($e$) in caption while ($e^{-}$) in the subfigure. ($\nu$) in caption while $\bar{v}$ in subfigure. $d$ in caption while $\bar{d}$ in subfigure. Moreover, subfigure (b) is too small to see the detail; the reviewer did not get the point of how it is related to your problem or the model. The font in this subfigure is too small to see. The review suggests presenting the subfigure (b) in a better way to give more insights into your problem or model.

2. There are several typos in the paper. For example, line 8, ``latent variation diffusion model``, is supposed to be ``latent variational diffusion models`` Line 192, the $\hat{\cdot}$ is on $z_t$ or $t$?

3. Some definitions are not clear. For example, how did you calculate $\hat{E}$ and $\hat{p}$ from the predicted value in Equation 10? What is the norm of the $\lambda_c|\cdot|$ in Equation 10? Is it $L_1$ norm or $L_2$ norm?

4. The author uses a deterministic encoder and claims it is better than the variational encoder. According to the author, the latter only has limited benefits while increasing training variance and complexity. However, there is no experiment result to support the claim. More ablation study is needed to justify the claim. Still, it sounds strange to the reviewer that the model is called a variational latent diffusion model but only uses a deterministic encoder.

5. The author proposed three variants of the model. Conditional, unconditional, and the last one, conditional encoder and unconditional decoder. The way of presenting the results confuses the reviewer. Firstly, which one do you promote to use in the conclusion? It seems in Table 1. VLD is better in the latter three metrics, while the UC-VLD is slightly better in the first three metrics. However, Figure 2 shows the framework of VLD, while all the Figures in the result part and the appendix part only show the results for the UC-VLD. If the authors want to promote VLD, consider adding relevant plot results for VLD. If the UC-VLD is better, consider changing Figure 2 to reflect this point.

6. A physics-informed consistency loss is proposed with a hyperparameter $\lambda_{c}$; how did you adjust the value of this hyperparameter? Moreover, what would be the effect of including and not including this loss?

7. The author claims the model is aimed at high-dimensional inverse problems. However, the training cost for $55$ dimensional variables is expensive, $24$ hours. Moreover, the designed latent space has a higher dimension than the input instead of the commonly used lower latent space dimension for compression. This design could introduce additional costs. And the reviewer is concerned with the scalability of the current model to higher dimensional problems.

8. What is the limitation of the current work?

**Limitations:**

The author did not mention the limitations explicitly. I would be curious what would be the limitation of the proposed model.

---

> ### Author Rebuttal · Authors · 2023-08-09
>
> We thank the reviewer for their insightful questions and comments.
>
> 1. We agree with the reviewer that Figure 1 could benefit from additional context. The intent of this figure was to offer a visual illustration of parton configurations and detector observations, as subsequent sections describe this data solely in terms of kinematic vectors. We have augmented the captions to clarify this intent.
>
> 2. We thank the reviewer for noticing these  detailed errors. We have performed a pass through the paper and fixed these errors along with other small wording changes. We have also added the missing Figure number for Figure 1 which did not render in our original submission.
>
> 3. As we train our model on the task of reconstruction, the inputs and outputs of our network consist of the parton momentum vectors, defined by the components $(M, \log E, p_x, p_y, p_z)$. These vectors represent specific physical quantities, such as mass and energy, and the network provides approximate values for these kinematic parameters. In response to this question and comments from other reviewers who requested clarification on the letter symbols and their associated physical quantities, we have added further details to this description. In relation to Equation 10, as all values are scalars, the bars represent a simple absolute value rather than a vector norm. We have clarified this with an explicit formulation in terms of mean absolute error.
>
> 4. We use a deterministic encoder only for the detector (conditioning) inputs, not the parton encoder. We have clarified this distinction in the text. We note in the text that we have the option to make the detector encoder variational, but we did not expect any benefit from this approach. Our decision aligns with previous work, including methodologies like CVAE and LDM, which similarly utilize deterministic encoders for their conditioning. This choice was also informed by preliminary experiments, which were insufficiently conclusive to include in our study. We have therefore also revised this section to more precisely articulate that we opted not to explore a variational encoder for the detector observations, owing to both a lack of compelling motivation and following existing precedent.
>
> 5. We thank the reviewer for identifying the discrepancy and the absence of detail in Figure 2, as well as for their insightful comment. We have revised Figure 2 to include annotations that highlight the differences between the three variations, thereby providing additional visual intuition to their definitions. We have included this updated figure in this response. We have also clarified that we view the UC-VLD as the best model for this dataset  due to its performance on the bin-independent metrics and present its results in the graphical comparisons.
>
> 6. The consistency loss scale functions as a hyper-parameter, comparable to the weight regularization scale, and requires empirical tuning. Our computational limitations precluded a comprehensive hyper-parameter sweep for this parameter, so we selected a conservative value of 0.1, guided by the loss magnitudes observed during training. We observed that omitting this loss leads to a broader mass reconstruction, failing to capture the sharp peak in the mass features, and this trend was consistent across all models. To ensure a fair comparison, we applied the same loss and loss scale for all models.
>
> 7. We are indeed looking into higher dimensional problems and problems where the data could have a variable number of dimensions. Diffusion models are infamously very computationally expensive to train, even within existing application domains. We note that our problem has 55 dense features, where every feature is approximately independent and meaningful; this is in contrast to images, which have much higher dimensionality, but each individual feature is less informative. Due to this difference, we opt to make our latent space with higher dimensionality than our data to allow the VAE to learn a fine-tuned latent space for the diffusion objective.
>
> 8. We address this question in the global rebuttal.

---

### Author Rebuttal · Authors · 2023-08-09

We thank all of the reviewers for their detailed comments and questions. We begin with a discussion of the limitations as requested by two of the reviewers and then answer the remaining questions in individual responses. We also include a document with updated figures.

Limitations
--------------
We note in the text that the experiments performed in this study are limited to parton unfolding of a specific event topology and on simulated data. The method is general and may be applied to arbitrary topologies, with our choice guided by the limits of current baseline methods. We think this exploration is robust as many practically explored topologies are simpler than our semi-leptonic $t\bar{t}$ test-case. We note in the text that the next step would be to perform particle-level unfolding which is not specific to a particular topology, and we believe this technique may be applied to this more general problem as well with little modification. Another common limitation of training on simulated data is the imperfection of the simulation and the possibility of skewing the results compared to real detector data. We note in the text that we must adjust the model’s simulation bias by employing real data, and we cite potential techniques, such the as iterative approach of ICINN, to accomplish this adjustment. We have added further discussion to the text to illustrate these limitations and future solutions.

---

### Decision · Program_Chairs · 2023-09-21

**Decision:**

Accept (poster)

**Comment:**

The reviewers were mixed on this paper and sought clarification on a number of points.  The authors engaged in the rebuttal process to provide answers to many of the questions raised.  While not all reviewers recommended acceptance, after reading the reviews, discussion and paper I recommend acceptance.  I believe the paper considers a challenging and interesting problem.  The proposed methodology appears to be well tailored to the task and demonstrates significant performance improvements over existing techniques.  The authors should revise the paper to consider the issues raised in the review and discussion process, including additional clarifications.